# Transmembrane Membrane Readers form a Novel Class of Proteins That Include Peripheral Phosphoinositide Recognition Domains and Viral Spikes

**DOI:** 10.3390/membranes12111161

**Published:** 2022-11-17

**Authors:** Michael Overduin, Anh Tran, Dominic M. Eekels, Finn Overduin, Troy A. Kervin

**Affiliations:** 1Department of Biochemistry, University of Alberta, Edmonton, AB T6G 2H7, Canada; 2SMALP Network, Edmonton, AB T6R 1B9, Canada; 3Institute of Nutritional Science, University of Potsdam, 14476 Potsdam, Germany

**Keywords:** C2 domain, FYVE domain, lipid recognition, peripheral membrane protein, PH domain, PX domain, SARS-CoV-2, sorting nexin, coronavirus spike, synaptotagmin, transmembrane protein

## Abstract

Membrane proteins are broadly classified as transmembrane (TM) or peripheral, with functions that pertain to only a single bilayer at a given time. Here, we explicate a class of proteins that contain both transmembrane and peripheral domains, which we dub transmembrane membrane readers (TMMRs). Their transmembrane and peripheral elements anchor them to one bilayer and reversibly attach them to another section of bilayer, respectively, positioning them to tether and fuse membranes while recognizing signals such as phosphoinositides (PIs) and modifying lipid chemistries in proximity to their transmembrane domains. Here, we analyze full-length models from AlphaFold2 and Rosetta, as well as structures from nuclear magnetic resonance (NMR) spectroscopy and X-ray crystallography, using the Membrane Optimal Docking Area (MODA) program to map their membrane-binding surfaces. Eukaryotic TMMRs include phospholipid-binding C1, C2, CRAL-TRIO, FYVE, GRAM, GTPase, MATH, PDZ, PH, PX, SMP, StART and WD domains within proteins including protrudin, sorting nexins and synaptotagmins. The spike proteins of SARS-CoV-2 as well as other viruses are also TMMRs, seeing as they are anchored into the viral membrane while mediating fusion with host cell membranes. As such, TMMRs have key roles in cell biology and membrane trafficking, and include drug targets for diseases such as COVID-19.

## 1. Introduction

Integral membrane proteins represent about a quarter of all proteins and the majority of drug targets [1]. Additionally, about 5% of all human proteins are membrane readers that contain folded domains capable of transiently recognizing phospholipids and especially PIs to mediate delivery to subcellular compartments [2]. Such peripheral membrane proteins include another ~30 drug targets [3]. Here, we illuminate the subset of human proteins that contain both a peripheral membrane domain able to bind specific lipid ligands and a TM helix that embeds into the bilayer. These are compared with the architecturally similar coronavirus spike proteins, which also span and bind membranes [4], in order to glean insights into their ability to enter host cells and exploit organelle surfaces.

For simplicity we focus here on representative members of each class of proteins that contain at least one of the ~70 known membrane reader domains [2], as well as a TM region. These proteins are referred to here as “transmembrane membrane readers” (TMMRs) to highlight their ability to stably anchor through a membrane-spanning structure while scanning for and recognizing lipids with a separate module. Their diverse mechanisms are contrasted with the spike protein, which binds host membranes via a multi-step mechanism that is mediated by its membrane-interacting sites in the N-terminal domain (NTD) and receptor binding domain (RBD) [4,5], while a C-terminal helix spans the viral envelope as well as intracellular compartments during virion assembly.

There are no 3D structures of full-length TMMRs embedded in a native lipid bilayer, as these complexes are difficult to study in their intact forms. Structures of such biological membrane:protein assemblies (“memteins” [6]) remain challenging to obtain as the constituent lipids of the bound asymmetric bilayer are typically scrambled and lost during purification. Their visualization benefits from the development of native nanodisc systems and high resolution structural biology methods including cryo-electron microscopy (cryo-EM) for ex vivo characterization of lipid-bound proteins [7]. In contrast, conventional nanodiscs formed using synthetic lipids and detergents to replace the asymmetric biological bilayer can lead to artifacts [6], and would disrupt the native conformation of a TMMR bound to two biological membranes. Resolution of TMMR structures informs our proposal that these machines are commonly found at membrane contact sites and have architectures that allow them to play roles in membrane trafficking and fusion. 

New computational tools allow modelling of TMMRs. Full-length protein structures can be predicted from programs including AlphaFold [8] and RoseTTAFold [9]. While such artificial intelligence and machine learning methods accurately predict structures of soluble proteins [10], TMMR protein structures are more challenging to predict due to their biphasic nature, flexibly linked domains, and multimeric states. Hence, we focus here on confidently predicted domains and compare models with high resolution experimental structures to define membrane-binding surfaces of TMMRs. The structures of linkers in these models are of low confidence and likely allow dynamic interactions between membranes. 

Several computational tools can be used to identify membrane binding residues in structures of TMMR domains. These programs include Positioning of Proteins in Membranes (PPM) [11,12], Ez-3D [13], and Membrane Optimal Docking Area (MODA) [14]. The latter uses an algorithm trained on validated phospholipid binding surfaces to score all the residues in protein structures including multimers for their membrane binding propensities. We previously applied this approach to discover validated lipid binding sites in bacterial and viral trafficking proteins [15,16], prions [17], SARS-CoV-2 spike proteins [4,5], and eukaryotic membrane readers [2,18]. Based on these experiences, we apply MODA here to identify most human TMMRs and their membrane-interaction surfaces, thus illuminating the relationship between their structure and function. Further investigation is warranted, as deducing the specificities and binding pockets of the multiple lipid ligands will require experimental characterization, as current algorithms cannot accurately predict such facets. 

Several recurrent themes emerge here for this new class of proteins. We find that TMMRs share features including (1) similar lipid specificities for each compartment, (2) enhanced affinity through multimerization and juxtaposed binding sites, (3) terminal placement of respective peripheral and TM domains, (4) endocytic targeting by pH-sensitive His-clusters, and (5) firm membrane anchoring by palmitoylation of sets of proximal cysteines or an adjacent series of TM and peripheral membrane domains. We propose that these features are critical for strong, selective, and dynamic membrane tethering for fusion and transport to occur between cells, organelles, and host membranes and viral particles.

## 2. Materials and Methods

### 2.1. Sequences and Modifications

Protein sequences were obtained from UniProt [19]. The superfamilies of membrane readers were surveyed for TM helices in the SMART tool [20]. Signal sequences were predicted using PrediSi [21] and SignalP [22]. The Group-based Prediction System (GPS) Lipid [23] and pCysMod tools were used to predict the presence of protein palmitoylation, myristoylation, and prenylation sites, and modified sites were also identified and confirmed from the published literature.

### 2.2. Protein Structures

Membrane protein structures were obtained by surveying the RCSB Protein DataBank (PDB) [24], Membranome [25], OPM [12], PDBTM [26], MemProtMD [27], and mpstruc [28]. Models of structures were generated using AlphaFold [8] and Rosetta [9]. The protein structures and lipid binding sites were visualized with PyMol [29].

### 2.3. Transmembrane Helix Prediction

The PolyPhobius [30] and HMMpTM [31] programs were used to predict the positions of TM helices using the protein sequences as input. The results were compared to the output of the SignalP, PrediSi and MODA programs, with the latter predicting membrane interacting positions within folded protein structures. This allowed elimination of false positives such as putative TM helices in the INPP4A and INPP4B proteins and at the N-terminus of Mep1a.

### 2.4. Coiled Coil Prediction

Coiled coils, which mediate oligomerization of proteins via two or more α helices, were predicted using Marcoil [32] in order to assess their presence in TMMRs as they are able to increase binding avidity for membranes. 

### 2.5. Membrane Docking Site Prediction

The MODA program was used to identify membrane-binding surfaces. This algorithm was trained on a family of well-characterized peripheral membrane proteins and assigns a membrane binding propensity to each residue based on its likelihood of interacting with phospholipid bilayers [14]. Missing coordinates in structure files were not modelled; instead, other similar structures were considered where gaps existed. Proximal sets of at least two residues that have MODA scores of 30 were ranked as having significant membrane binding propensities. The output from MODA was analyzed in Excel (Microsoft) to assess consistency across homologous structures and to generate heatmaps showing consensus membrane binding profiles. Membrane-interacting surfaces in peripheral membrane domains were confirmed using PPM 3.0 [11,12], which predicts positions of protein structures on fluid anisotropic solvent slabs that resemble membranes. The consensus TM helices and membrane docking sites were depicted in images produced in PyMol and ICM Browser.

## 3. Results

Analysis of protein structure databases reveals a diversity of membrane proteins containing both a TM helix and at least one peripheral membrane domain that can recognize one or more of the eight different PI lipid headgroups that mark various subcellular compartments (Figure 1). In particular, we found 54 distinct human proteins that contain both TM helices and at least one canonical membrane reader domain. Together, they represent 2.8% of all membrane readers identified to date [2], and only 0.3% of all human proteins, perhaps explaining why this functional superset has not yet been reported to our knowledge. The caveats are that the lipid binding properties of many of these TMMR proteins have not yet been experimentally characterized, and more membrane readers are likely to exist that recognize the thousands of other phospholipids and glycolipids.

Our investigation reveals that TMMRs contain a surprisingly wide variety of peripheral membrane domains and can recognize an array of PIs found in eukaryotic organelles. Of these, the C2 domain is the most prolific, being found in 89 copies in 28 different human TMMRs. The next most prevalent are GRAM and SMP domains (present in 6 proteins), PH domains (5 proteins), GTPase, PDZ and PX domains (4 proteins each), and C1, RGS and StART domains (3 proteins each), while CRAL-TRIO, FYVE, MATH, and WD domains are found in single TMMRs in humans. These domains represent 14 distinct folds capable of reading and identifying lipid molecules in bilayer surfaces while being simultaneously anchored by transmembrane elements. Membrane-spanning helices are found singly in the N- and C-terminal regions of 21 and 22 TMMRs, respectively, as pairs at the N- and C-termini of 7 and 2 TMMRs, and quadruply in GRAMD4 and protrudin, indicating a rich architectural diversity responsible for membrane-membrane attachment in organelle-dense eukaryotic cells.

Below we focus on the best-characterized human representatives of each TMMR family. The multidomain proteins we consider here are stably anchored into and localized in membranes throughout the insides of human cells where they typically recognize PIs to initiate localized trafficking and signaling events (Table 1). In contrast, the viral spike protein mediates entry to mammalian cells via endocytic and plasma membranes and then guides viral assembly on the ER-Golgi intermediate compartment (ERGIC) and egress through exocytic vesicles via a multifaceted head [4,5]. Hence, although there may be instructive parallels, polyfunctional spikes are likely to possess additional features not found in eukaryotic TMMRs. 

### 3.1. The C2 Domains Are the Most Common TMMR Module

The C2 domain superfamily represents the most extensive group of TMMRs, with the synaptotagmins forming the largest subset. Synaptotagmin proteins mediate calcium-dependent fusion of neuronal membranes at synapses in response to action potentials [88]. They are composed of two C2 domains that follow a N-terminal TM helix, which is bordered by palmitoylated cysteine residues (except in Syt12). These proteins form homo- and hetero-oligomeric rings, and recognize specific lipids as well as SNARE (soluble N-ethylmaleimide sensitive factor attachment) proteins that zipper into four helix bundles to provide energy for the fusion reaction. Of all the synaptotagmin isoforms, Syt1 is the best characterized, including how it mediates synaptic vesicle docking, priming, and calcium-triggered fusion. The structure of its tandem C2 domains bound to a SNARE and membrane has been solved by cryo-EM [97]. Both C2 domains insert loop 3 including bound calcium ions into the membrane surface, while their loop 1 and KK-327 elements project towards the membrane where they can engage phosphatidylserine (PtdSer) and phosphatidylinositol 4,5-bisphosphate (PI(4,5)P_2_) ligands [90,91]. The second C2 domain helps hold the plasma and vesicle membrane apart through its complex with palmitoylated SNARE proteins. Our MODA analysis of the structural models derived from Rosetta and AlphaFold2 identify additional membrane-binding elements including the TVATVL-22 and ELHKIPLP-57 sequences which are N-terminal to the TM helix (Figure 2), and five palmitoylated cysteines are found in a helix that may convert to β strand conformation when bound to a lipid bilayer [98]. Together these elements contribute to the formation of oligomeric assemblies on membranes that are primed for calcium-dependent fusion and neurotransitter release.

The extended synaptotagmin (ESyt) proteins serve to connect the endoplasmic reticulum (ER) and plasma membrane (PM) and are found throughout eukaryotes. Their N-terminal regions anchor into the PM while their series of C2 domains mediate Ca^2+^-dependent PI(4,5)P_2_ recognition [41]. The long hydrophobic grooves of their intervening SMP domains interact with glycerophospholipids [43], and shuttle lipid molecules between membranes.

The C2CD2L protein, also known as TMEM24, contains an N-terminal TM helix followed by PI-binding SMP and C2 domains that mediate membrane interactions and replenish PI(4,5)P_2_ levels in the PM [33]. Its C-terminal polybasic region also mediates PM binding, unless negatively regulated by calcium-dependent phosphorylation of serine residues not found in its paralogue C2CD2 [34].

A pair of human proteins termed MCTP for “multiple C2 domain and transmembrane region proteins” contain three C2 domains as well as C-terminal TM domains and palmitoylation sites. They are anchored into the ER and promote the release of synaptic vesicles [54]. While the C2 domains bind calcium and possess lipid binding signature sequences, their ability to bind membranes has not yet been demonstrated [52,53].

Finally, six ferlin proteins named DYSF, Fer1L4, Fer1L5, Fer1L6, MYOF and OTOF contain six or seven C2 domains. Binding studies have established the phosphoinositide and calcium interactions of the C2 domains of DYSF [38] and OTOF [67]. Those of DYSF are known to mediate protein dimerization [40] and play roles in Ca^2+^ signaling and sarcolemma membrane repair [99]. These proteins also contain four helix bundles known as FerA domains that also bind phospholipids in a calcium-dependent manner [100]. Together these mediate regulated fusion events between the PM and late-endosomes or trans-Golgi network and recycling endosomes [39].

### 3.2. C1 Domain-Containing Lipid Kinase

Like C2 domains, the C1 domains are named for their discovery as conserved regions in Protein Kinase C, but are only found in two TMMRs. Of the ten human diacylglycerol (DAG) kinases, the epsilon isoform is distinguished by a pair of C1 domains and the presence of an N-terminal TM domain, which help anchor it to the ER and PM [37] and contribute to DAG lipid binding and fatty acyl specificity [101]. The full-lengh DGKε structure predicted by AlphaFold 2 includes a TM helix that protrudes away from the C1 domains, the second of which is packed against the conjoined catalytic and accessory domains (Figure 3). Our MODA analysis shows that the first C1 domain projects membrane-interactive motifs including DL-67, SQP-71 and LQ-83 towards the TM domain in the AlphaFold2 model, and Rosetta also predicts F68 to be membrane-interactive. Together these elements form a binding cleft that could provide lipid substrate specificity. This uniquely membrane-bound DGK family member is regulated by highly curved membranes, its N-terminal segment [35], and its product phosphatidic acid (PA) [36]. In contrast to canonical TMMRs, the function of DGKε is to carry out ATP-dependent phosphorylation of diacylglycerol to form PA. This enzyme specifically accommodates substrates composed of 1-stearoyl-2-arachidonoyl acyl chains, which are then incorporated into PI lipids. 

### 3.3. PDZ Domain-Containing Integral Membrane Proteins

The PDZ domain is found in 149 human proteins including four TMMRs. These modules typically mediate interactions with C-terminal receptor sequences while a third of PDZ domains potentially bind PI lipids, typically in the PM [102]. The PDZD8 protein localizes to the ER and contacts late endosomes and lysosomes, and acts as a lipid transfer protein that replenishes PI(4,5)P_2_ levels [68]. It is composed of an N-terminal TM helix, SMP, PDZ, C1, and C-terminal coiled-coil domains. The structure of its coiled coil bound to Rab7-GTP complex has been determined [103], suggesting how this complex assembles on late endosomal membranes via the C1 domain and palmitoylated Rab7 partner to transfer lipids from the ER membrane. Our MODA analysis indicates that the SYJ2B protein, which contains a C-terminal TM domain and a single N-terminal PDZ domain, lacks obvious lipid binding signatures in the three available structures (PDB: 2ENO, 2JIK, 2JIN). In contrast, our MODA analysis predicts that the 1.46 Å resolution crystal structure of the LIM only protein 7 (LMO7) PDZ domain solved by S. Yokoyama’s group contains a major phospholipid bilayer docking site, which is only partially apparent in the AlphaFold2 model (Figure 4). The lipid specificities of the PDZ domains of PDZD8, LMO7 [51] and the protease Htra2 [50] remain unclear. However, PDZ-lipid interactions could contribute to the localization of LMO7 to the PM and nuclear envelop, where it plays a role in cell differentiation.

### 3.4. PH and GRAM Domains Are Common in TMMRs

PH domains comprise the largest superfamily of PI-binding modules, and are structurally related to GRAM domains. The lipid transfer proteins ORP5 and ORP8 bind PI4P as well as PtdSer via their membrane binding PH domains (Figure 5) and are anchored into ER membranes by C-terminal TM helices, while their oxysterol-binding protein (OSBP) domains bind PIs and sterols [62]. Structural and biophysical analysis shows that these PH domains bind to multiply phosphorylated PIs, with R201Q and R158Q mutations in ORP8 abolishing PI binding, and corresponding mutations in ORP5 blocking localization to ER-PM junctions [63]. These lipid transfer proteins also appear to be involved in the formation of lipid droplets, which serve as energy reservoirs, at ER-mitochondria contact sites [64,65].

Single PH domains are also found at the N-termini of the short PLEKHB1 and PLEKHB2 proteins (otherwise known as Evectins), which contain a C-terminal TM domain that inserts into the Golgi membrane [70]. The PLEKHB2 PH domain binds PtdSer (but not PI lipids) via residues including R11, R18 and K20, and mutation of these positions compromises the protein’s membrane localization [72]. Although structures of TMMR PH domains bound to lipids or bilayers are not available, those of their relatives are available (Figure 6). All PH domains fold into a 7-stranded antiparallel β-sheet sandwich with one or two C-terminal α-helices. Membrane insertion is mediated by the β1-β2 loop and PIs are bound via a KX_n_(K/R)XR binding motif [104]. The PI ligand is typically recognized by the closed side of this loop, as in the case of the FAPP1 PH domain, which specifically recognizes disordered PI4P-containing bilayers and induces tubule formation in membranes [105]. The ASAP1 PH domain recognizes PI(4,5)P_2_ as well as a second lipid molecule, providing additional membrane anchoring [106].

Both PH and SMP domains as well as a pair of TM domains are found in the TEX2 protein. It localizes to ER-late endosome contact sites and in conjunction with PDZ8 suppresses endosomal PI(4,5)P_2_ levels [68]. The yeast homolog of TEX2 locallizes to contacts between the ER and vacuole membrane and upon stimulation, to ER-Golgi interfaces to facilitate lipid transport [95,96], suggesting a conserved function.

The GRAMD family or proteins typically contain C-terminal TM helices that anchor them to ER membranes in addition to an N-terminal GRAM domain. The GRAMD1 proteins also contain an intervening StART domain. The exception to this pattern is GRAMD4, which contains 4 predicted TM helices followed by a single C-terminal GRAM domain. These oligomeric proteins are thought to be involved in the vesicle-independent movement of cholesterol between compartments to the ER [44]. Their StART domains bind to cholesterol [46], as resolved by the GRAM1DC complex [45], while their GRAM domains bind synergistically to cholesterol and anionic lipids such as PIs or PtdSer [47]. The GRAM domain of the corresponding yeast protein, Lam6, lacks the cluster of basic residues that typically allows PH domains to recognize PIs and instead offers a more hydrophobic surface [107], suggesting a novel recognition mode. The coincident interactions of these multiple lipid ligands tether the PM and ER membranes, and allow redistribution of lipid molecules between the juxtaposed compartments.

### 3.5. Sorting Nexins and PX Proteins

The Phox Homology (PX) domain is the signature module of the sorting nexin protein family and is the best-characterized fold capable of recognizing all phosphoinositides. A subfamily of four sorting nexins (SNX13, SNX14, SNX19, and SNX25) all include a series of cytosolic domains and are anchored to the ER by TM helices at either terminus. The PX domains of Snx13 and Snx19 are known to bind PI3P, thereby tethering endolysosomes (ELs) or lipid droplets (LD) to the ER [79]. The SNX13 protein structure is modelled in Figure 7 and is homologous to the yeast protein Mdmp1, which similarly tethers vacuoles to the ER [80], indicating a conserved function. SNX19 is the only member of this subfamily that does not contain an RGS domain, a module which negatively regulates G protein coupled receptor (GPCR) signaling by stimulating the hydrolysis of GTP to GDP in Gα subunits. These proteins exhibit various GPCR interactions; the PX domains of SNX13 interacts with Gαs [108], while PI binding by the PX domain is required for SNX14′s interaction with serotonin receptor 5HT6 [109], and SNX19 associates with the dopamine receptor D1R [110]. Such interactions may be regulated by PI(3,4,5)P_3_ interactions of RGS domains [111].

The mechanism of membrane insertion by PX domains is understood based on 3D structures and docking studies. The PX fold comprises an antiparallel β sheet which packs against a helical bundle, with the termini projecting away from the membrane-binding surface. Although the structures of Snx TMMRs have not been determined and their models contain uncertainties (Figure 7), their lipid recognition sites are likely similar to those of other PX domains, with PI complexes resolved for Grd19, p40^phox^, p47^phox^, Snx3, Snx9, Snx11 and Vam7 proteins [112,113,114,115,116,117,118,119,120]. A set of basic and hydrophobic residues contact the PI ligand and insert into the membrane, as seen with Snx3 (Figure 8). Membrane insertion is mediated by a hairpin loop between the β1 and β2 strands, the β3-α1 junction and an element linking the proline-rich element and α2 helix [116]. The PI3P lipid is bound by the shared R[Y/F]X_23–30_KX_13–23_R motif, where X is any residue. The first Arg residue interacts with the 3-phosphate, the aromatic group packs against the inositol ring, the Lys residue contacts with the 1-phosphate and the second Arg residue hydrogen bonds with the inositol’s 4- and 5-hydroxy groups. The PX domain of Snx11 recognizes PI(3,5)P_2_ in a similar manner, with a conserved lysine engaging the 5-phosphate [120]. Additional lipids including phosphatidic acid (PA) or PtdSer can be accommodated in the membrane docking surface as seen in structures of the PI(3,4)P_2_ -complexed PX domain of p47^phox^ [114]. The dipole formed by the PX domain provides attraction to negatively charged membrane surfaces, while aliphatic and aromatic groups anchor into the lipid bilayer as elucidated for Snx3 [116]. Sorting nexins use this mechanism to assemble retromer complexes on curved membranes for attachment of cargo [121]. 

### 3.6. FYVE Protein-Containing Protrudin

Protrudin regulates the growth of neurites by directing traffic at endosomes within neuronal membranes. It contains a FYVE domain, which interacts with several PIs, as well as a coiled coil and RAB GTPase interactions [76,77]. Protrudin’s four predicted TM domains insert into the ER where it forms membrane contact sites with late endosomes. Protrudin extracts lipids from these compartments [69], and engages partners including PDZ8, VAP [122], and KIF5 [123] to assist in trafficking at endosomes to ensure neuronal polarity and integrity.

### 3.7. Sec14 Domain-Containing MSPD2 Protein

The motile sperm domain-containing protein 2 (MSPD2) is found at membrane contact sites of the ER, in which it is anchored by its C-terminal TM helix. It engages lipid droplets via its CRAL-TRIO domain, which specifically recognizes packing defects and negatively charged phospholipids [60]. This module is also referred to as the SEC14 domain, and is known to bind PI lipids [124,125]. MSPD2 also contains a major sperm protein (MSP) domain which is exposed to the cytosol and binds FFAT motifs found in protein partners that facilitate contacts with endosomes, Golgi and mitochondria [61].

### 3.8. WD Repeat-Containing PREB Protein

There are 169 proteins encoded by the human genome that contain Trp-Asp (WD) repeats domains. Peripheral membrane proteins such as Coronin 1A (COR1A) bind to PI(4,5)P_2_ in membranes through their WD repeats and α helical extension, and plays roles in the regulation of phagosome formation [126], actin filament disassembly and cytoskeletal reorganization [127]. Other members of this superfamily such as the prolactin regulatory element-binding protein (PREB) contain a TM domain and function as transcription factors. Crystal structures of the cytoplasmic domain of Sec12, which is the yeast homolog of PREB, reveal how the ectodomain binds to a Sar1, a small Ras-like guanosine triphosphatase (GTPase) of the Arf family that inserts into the bilayer [128]. This complex regulates the formation of a vesicular intermediate in protein transport from the ER to the Golgi apparatus. A basic surface of PREB remains membrane-accessible in the complex. Our MODA analysis of the PREB structure reveals that part of the β propeller fold and segments preceding the C-terminal TM helix engage the membrane (Figure 9), which may induce membrane curvature and fission. Such trafficking events are negatively regulated by kinases including LTK, which resides in the ER where it associates with PREB [74].

### 3.9. GTPase Domain-Containing Miro Proteins

Mitochondrial trafficking is mediated by the Miro1 and Miro2 proteins, which are inserted into mitochondrial outer membranes by their C-terminal TM helices [57]. They both contain two calcium-binding EF-hand domains that are flanked by a GTPase domain, which appears to mediate dimer formation [58,129]. Their membrane binding mechanisms are unknown, and their sequences diverge from GTPases including K-Ras which are known to bind PI(4,5)P_2_-containing liposomes [130] and to dimerize [131]. Miro proteins form clusters at contact sites linking the ER and mitochondia to direct membrane transport [132], and also localize to peroxisomes, where they can mediate membrane fission [56].

### 3.10. MATH Domain-Containing Proteins

The membrane interactions of several other proteins that contain TM helices and modules known to function as membrane readers are less well characterized. Examples include the metalloprotease Mep1A, which forms oligomers and ring-like assemblies [55] and also contains a MATH domain that is known to bind PIs in other proteins including TRAF4 [133]. This meprin along with its partner Mep1B are concentrated in kidney and intestinal brush border membranes, from which they can be cleaved proteolytically for secretion [134]. A number of mutations in Mep1A including in its MATH domain are linked to inflammatory bowel disease [135].

### 3.11. Spike Proteins

Entry into host cells by the SARS-CoV-2 virus is mediated by the spike protein trimer. This homotrimer contains three topological domains. They anchor into the viral membrane via a C-terminal TM helix and ten palmitoylated cysteine residues while the rest of the protein projects away from the bilayer. The extracellular domain of Spike has two distinct subunits, S1 and S2, which perform different stages of the host–virus membrane fusion process. Initiating this process is the S1 subunit. In addition to binding the angiotensin converting enzyme-2 (ACE2) host receptor, S1 interacts with cell membranes via numerous membrane binding sites within the NTD and RBD of the exposed head [4,5]. Together they present a large flat membrane-binding surface which allows the virion to tether itself to the host cell membrane and close the distance between the viral and host membranes in preparation for ACE2 docking and membrane fusion. Analogous mechanisms may be employed for membrane interactions by viruses including Ebola, HIV, MERS and Zika, suggesting that TMMRs may be important targets for a range of epidemics [136].

## 4. Discussion

A diverse family of proteins that generally mediate connections between pairs of membranes is discussed here, which are characterized by the inclusion of both transmembrane and peripheral membrane domains capable of specific lipid recognition. Many members of this superset congregate at membrane contact sites that mediate trafficking between subcellular compartments. Such contacts are typically 10–30 nm, and can lead to approaches of under 2 nm that enable spontaneous fusion [75]. We propose that the general function of most TMMRs is to facilitate sufficiently close membrane juxtapositions to allow directed membrane fusion and transfer reactions. The preponderance of C2 modules indicate evolutionary divergence of TMMRs in eukaryotes, while the structural variety across the TMMRs suggests convergence on key membrane positioning functions for different organelles. This evolutionary pattern is highlighted by the SARS-CoV-2 spike protein, which rapidly acquired spike mutations that yield superior membrane binding propensities, culminating in the highly transmissible Omicron BA.1 variant [5]. The structure and function of this spike protein is similar to that of the syncytins, which are human fusogens acquired from viruses that have recently been resolved and also bind phospholipids [137].

Proteins that simultaneously interact with two membranes are already a familiar concept, as seen in the eukaryotic SNAREs and coronavirus spike proteins. The stepwise process by which these proteins mediate membrane fusion relies on the placement of their TM elements in bilayers and by attachment to distal membranes through multi-step mechanisms as detailed for SNAREs [90,91,97,98] as well as spikes prior to fusion [5]. Various models for translation of lipids between contact sites have been proposed. According to the tunnel model, the space between membranes is bridged by lipid binding domains that form a channel for moving lipid molecules [43]. In contrast, the shuttle model is based on a peripheral membrane domain docking to one membrane, which picks up lipids and then diffuses to another membrane to release this ligand [138]. The TM domain may play an important role in fusion through the lipid splay hypothesis, with lipid acyl chains fleetingly moving to the hydrophilic surface of membranes to promote fusion with other membranes when promoted by conformationally flexible TM domains that unlock the acyl chains from the membrane interior [139]. A lipid-binding domain on the same protein could then conceivably loosen lipids on the target membrane while holding the two compartments together to promote fusion. While many intermediates of fusion reactions remain to be characterized, the components and principles are emerging and lay the foundation for further analysis.

A recurrent, if not universal, feature of TMMRs is that they form multimeric assemblies on membranes. Half have been characterized experimentally as being dimeric, oligomeric or forming ring-like structures (Table 1). Another eleven of those listed here contain coiled coil regions that mediate multimerization. Most TMMRs present multiple membrane binding modules, particularly in the case of C2 domain proteins, and some of these presumably offer enhanced avidity for organelle surfaces. Twenty percent of TMMRs contain multiple transmembrane domains, and the MCTP, synaptojanin, and spike proteins contain palmitoylated regions near their TM domains, both of which further stabilize membrane anchoring. Thus, these proteins exhibit multivalent binding of several lipids, providing the avidity needed to stably connect organelle surfaces during membrane fusion.

The lipid microenvironment is a key component of membrane fusion and exchange reactions. Here, we focused on the human proteins that mediate heterotypic fusion between differing membrane compartments, as the mechanisms driving homotypic fusion within the ER and mitochondrial compartments are less well understood. Most TMMRs also partner with other proteins; for example, SNAREs help drive the fusion reaction between synaptic vesicles and the PM and may offer additional lipid interactions. Some lipids such as diacylglycerol and cholesterol induce negative membrane curvatures that promote fusion [140], with the latter also influencing the order in lipid bilayers and protein oligomerization. The negative charge of the membrane is critical as this generates a repulsive force between the proximal lipid bilayers until positively charged calcium ions or basic residues bind. Hence, acidic PtdSer and PI lipids play a role in maintaining membrane separation as well as serving as specific ligands for TMMR domains that allow electrostatic attraction when bound. Such charges are affected by cellular location, with FYVE domains binding PIs through a histidine-rich pocket in a pH-dependent manner to provide selectivity for acidic endosomal destinations [141]. By analogy spike proteins also exhibit pH-dependent conformations, and their membrane binding surfaces contain a cluster of His residues that may target host membranes in a pH-sensitive manner [4]. Further experimental investigation is needed to determine the specificities and binding pockets of the multiple lipid ligands, as current algorithms cannot currently predict this.

There are likely many more TMMRs that read lipids and anchor into juxtaposed membranes than the canonical membrane readers identified here. Several syntaxin proteins contain C-terminal TM helices bordered by juxtamembrane regions that are palmitoylated and PI(4,5)P_2_-binding and mediate calcium-dependent clustering on the PM to execute vesicle fusion [142,143]. There are four human junctophilin proteins that insert into the ER and sarcoplasmic reticulum membranes via C-terminal TM helices. They also contain a MORN motif domain that recognizes phosphatidic acid, PI4P and PI(4,5)P_2_ lipids [144] as well as phosphoinositide tails [145]. The yeast protein Ist2 interacts with PM lipids including PI(4,5)P_2_ via its N-terminal region while inserting into the ER via its C-terminal TM domain [146,147]. The related ER-resident family of ten human TMEM16 ion channels are modulated by lipids including PIs [148], which bind to cytosolic sites [149] with lipids also occupying a groove in their helical TM domain [150]. Such proteins may couple lipid scrambling and ion channel conductance functions. In addition, bacteria and enveloped viruses employ a large variety of proteins to fuse with host cell membranes. Hence, the cases discussed here represent a subset of the broader universe of TMMRs, underscoring their wider roles in cellular organization, entry and navigation. 

While TMMRs are presented here as a distinct class of proteins, we propose that many, if not most other membrane-spanning proteins also contain peripheral membrane elements. As such, “transmembrane protein” can be a misleading classification given that their peripheral membrane domains may be more important functionally. To illustrate, some synaptotagmins such as Syt17 mediate intracellular traffic yet lack TM helices while retaining the C2 domains and palmitoylated cysteines that are signatures of this family [88]. Hence, the peripheral membrane interactions of TMMRs may be most critical for understanding the mechanisms of how organelle-organelle interactions, trafficking in eukaryotic cells and invasion by enveloped pathogens are mediated.

## Figures and Tables

**Figure 1 membranes-12-01161-f001:**
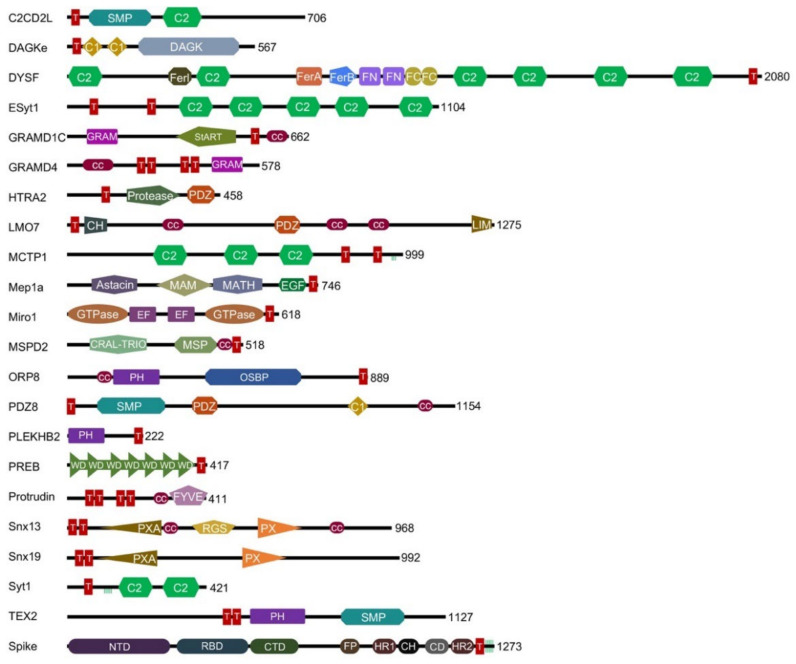
Domain architectures of TMMR proteins. Selected representatives of the diverse families of human TMMRs as well as the SARS-CoV-2 spike protein are shown. Each protein is labelled with its name, structural domains, and length. TM regions are depicted as red rectangles containing a “T”. Palmitoylated cysteines are shown as short vertical green lines.

**Figure 2 membranes-12-01161-f002:**
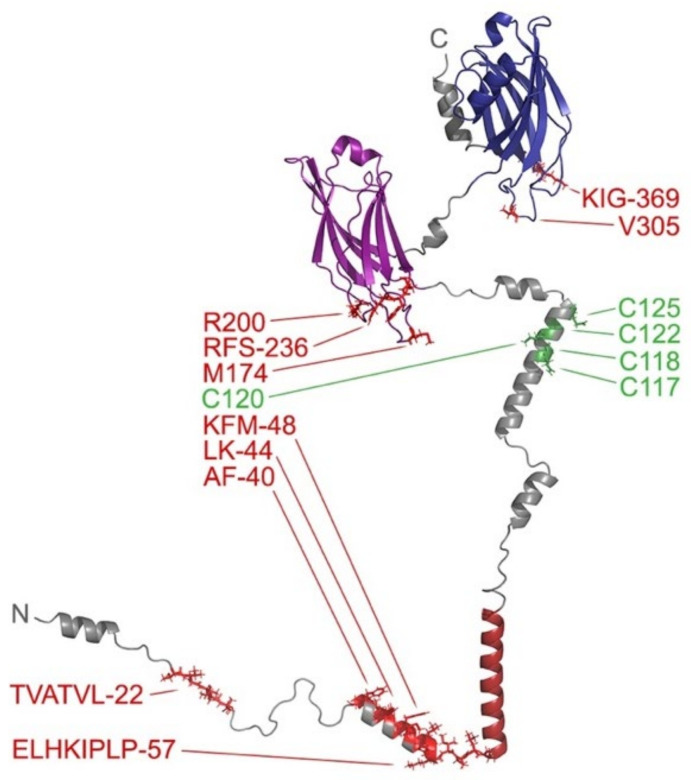
Model of full length Syt1 showing its membrane interactive elements. The TM helix and first and second C2 domains are colored dark red, purple and blue, respectively, in the Rosetta-derived structure. The rest of the backbone ribbon is gray. The residues in the peripheral elements with scores of at least 30 are labelled and in red with sidechains shown. The cysteines that are predicted to be palmitoylated are shown in green and labelled. The N- and C-termini are labelled. This model contains uncertain aspects including the positioning of the two C2 domain lipid binding sites and palmitoylated cysteines, which would normally engage membrane surfaces.

**Figure 3 membranes-12-01161-f003:**
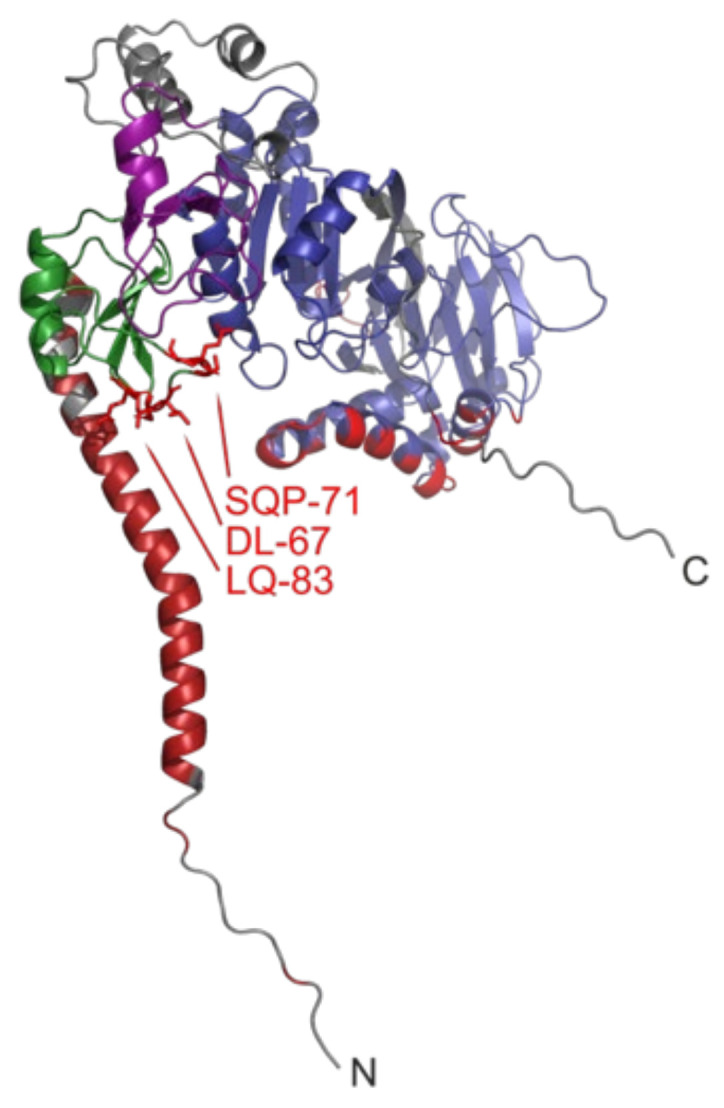
Structure of DGKε showing its membrane interactive elements. The TM helix, first and second C1 domains and catalytic domain of the AlphaFold2 model are colored dark red, green, magenta and blue, respectively. The rest of the backbone ribbon is gray. The residues in the peripheral domains with scores of at least 30 are red, with the sidechains of those of the first C1 domain drawn as sticks and labelled. The N- and C-termini are labelled.

**Figure 4 membranes-12-01161-f004:**
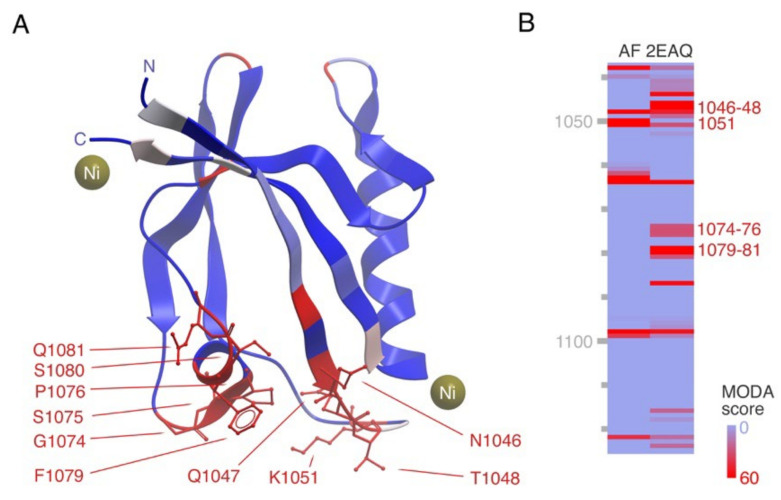
Structure of PDZ domain of LMO7 and its membrane binding site. (**A**) The crystal structure of the LMO7 PDZ domain (PDB: 2EAQ) is shown with the residues showing significant phospholipid bilayer binding propensities indicated in red, along with sidechains for those forming the major membrane binding surface. The N- and C- termini and two nickel ions bound in the crystal structure are also depicted. (**B**) The heatmap shows the residues with significant membrane binding propensities in the AlphaFold and crystal structures in red, as calculated by MODA, with the remainder in blue, based on the inset gradient.

**Figure 5 membranes-12-01161-f005:**
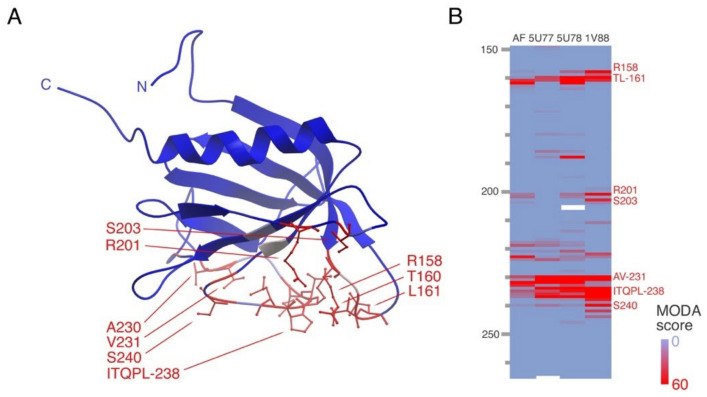
Structures and membrane binding surface of the ORP8 PH domain. (**A**) The structure (PDB: 1V88), which was determined by NMR by S. Yokoyama’s group, is shown with the residues showing significant phospholipid bilayer binding propensities indicated in red, along with sidechains for those forming the major membrane binding surface. The N- and C-termini are labelled. (**B**) The heatmap shows residues in shades of red for significant membrane binding propensities in the AlphaFold2 model, NMR and crystal structures, with the remainder in blue, based on the inset gradient. White cells represent unresolved residues in the structure.

**Figure 6 membranes-12-01161-f006:**
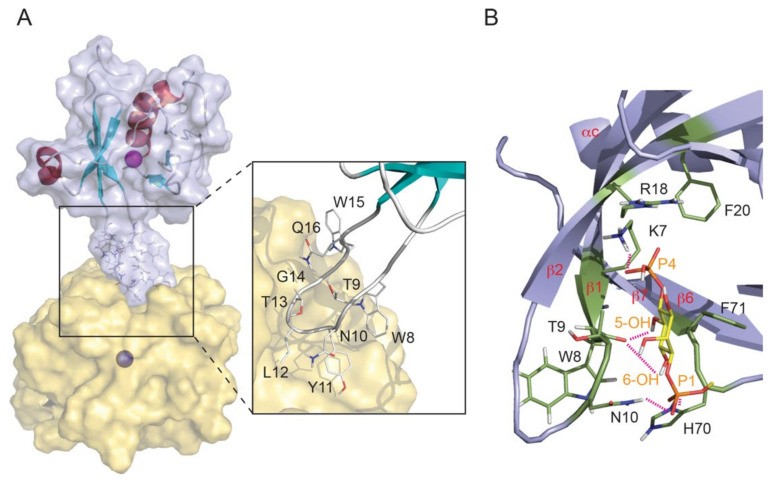
Structure of the PH domain of FAPP1 docked to a PI4P-containing micelle. (**A**) The backbone of the structure (PDB: 2MDX) is grey, helices and β-strands are shown in aqua and red, respectively, under a translucent silver molecular surface and the dodecylphosphocholine micelle is gold. The protein and micelle centres are marked with magenta and blue dots, respectively, and 40 Å apart. The β1–β2 loop residues that insert into the micelle are labelled. (**B**) The bound inositol ring and 1- and 4-phosphates are colored yellow and orange, respectively. Intermolecular hydrogen bonds are depicted as dashed lines to the sidechains of the labelled residues. Used by permission from [105].

**Figure 7 membranes-12-01161-f007:**
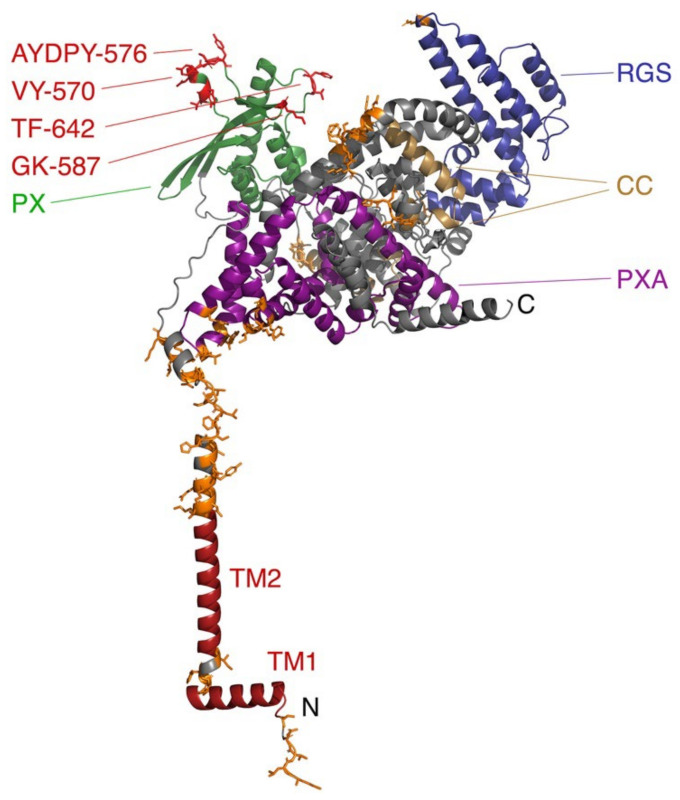
Structure of full length Snx13 showing its membrane interactive elements. The two TM helices, coiled coils, PXA, RGS, and PX domains of the AlphaFold2 model are colored dark red, sand, magenta, blue, and green, respectively. The rest of the backbone ribbon is gray. The sidechains of residues with membrane binding propensity scores of at least 30 in the PX domain and other peripheral elements are colored red and orange, respectively, while only those in the PX domain are labelled. The N- and C-termini are indicated. This model contains uncertain aspects including the perpendicular positioning of two TM helices, the inclusion of the two predicted coiled coils in a larger all-helical domain, and the orientation of the multitude of peripheral membrane docking elements including those of the PX domain away from each other and the TM helices.

**Figure 8 membranes-12-01161-f008:**
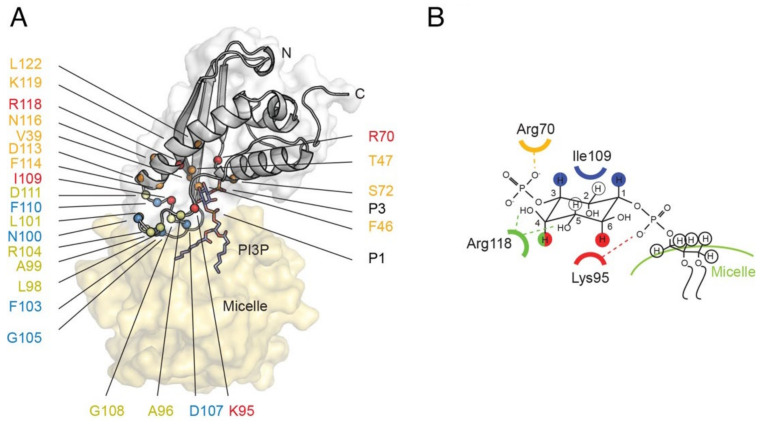
Structure of the Snx3 PX domain docked to a PI3P-contain micelles. (**A**) The Snx3 backbone (PDB: 2MXC) is grey with DPC micelle shown as a yellow surface; key residues are labelled and interactions with the PI3P headgroup are shown in (**B**). Residues which exhibit significant chemical shift perturbations upon interaction with the PI3P headgroup or micelles are coloured orange or yellow, respectively, while those exhibiting intermolecular interactions with PI3P or DPC are also displayed in red and blue, respectively [116] The phosphate groups of PI3P are indicated by P1 and P3, and hydrogen bonds are shown with dotted lines. Used by permission from [116].

**Figure 9 membranes-12-01161-f009:**
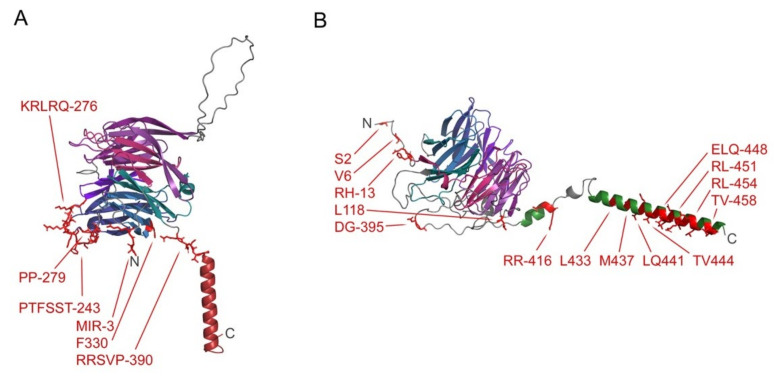
Structural models of WD repeat proteins showing their membrane docking sites. The PREB (**A**) and COR1A (**B**) structural models from AlphaFold2 and Rosetta, respectively, are shown with the seven WD repeats of their β propeller domains colored pink, violet, medium and dark purple and blue, and teal. The TM helix of PREB is vertically oriented dark red, while the membrane-interactive helical extension (HE) in COR1A is green and roughly horizontal. The rest of the backbone ribbon residues are gray. The peripheral residues displaying significant membrane binding propensities are drawn in red as sticks and labelled. The N- and C-termini are labelled.

**Table 1 membranes-12-01161-t001:** Human proteins containing both membrane reader domains and TM helices. The number and N- or C-terminal position of TM helices are indicated as are the names of membrane reader (MR) domains. Additionally, indicated are the known lipid ligands, oligomeric state or, if unknown, the presence of a coiled coil (cc). Cellular functions and locations are stated, including AP, autophagosome; EC, extracellular; EL, endolysosome; E, endosome; ER, endoplasmic reticulum; G, Golgi; LD, lipid droplet; LE, late endosome; LY, lysosome; MOM, mitochondrial outer membrane; NE, nuclear envelop; PM, plasma membrane; PO, peroxisome; PRCO, postsynaptic receptor-containing organelle; RE, recycling endosome; SDCV, secretory/large dense core vesicle; SV, secretory vesicle; TGN, trans-Golgi network; TV, trafficking vesicle; VM, viral membrane, along with relevant references.

Protein	Uniprot	TM	MR	Lipid Ligands	Oligomer	Function	Location	References
C2CD2	Q9Y426	1α-N	SMP, C2	PI		lipid transfer	ER	[33]
C2CD2L	O14523	1α-N	SMP, C2	PI		lipid transfer	ER, PM	[34]
DGKε	P52429	1α-N	C1 (2)	DAG		lipid kinase	ER, PM	[35,36,37]
DYSF	O75923	1α-C	C2 (7)	PtdSer, PI4P, PI(4,5)P_2_ (Ca^2+^)	dimer		PM, LE, LY	[38,39,40]
ESyt1	Q9BSJ8	2α-N	SMP, C2 (5)	PI(4,5)P_2_ (Ca^2+^)	dimer	lipid transfer	ER, PM	[41,42]
ESyt2	A0FGR8	2α-N	SMP, C2 (3)	PI(4,5)P_2_ (Ca^2+^)	dimer	lipid transfer	ER, PM	[41,43]
ESyt3	A0FGR9	2α-N	SMP, C2 (3)	PI(4,5)P_2_ (Ca^2+^)	dimer	lipid transfer	ER, PM	[41]
FR1L4	A9Z1Z3	1α-C	C2 (6)		cc			
FR1L5	A0AVI2	1α-C	C2 (7)					
FR1L6	Q2WGJ9	1α-C	C2 (6)		cc		PM, TGN	[39,44]
GRAMD1A	Q96CP6	1α-C	GRAM, StART	cholesterol	oligomer	lipid transfer	AP, ER, PM	[44,45,46]
GRAMD1B	Q3KR37	1α-C	GRAM, StART	PtdSer, cholesterol	oligomer	lipid transfer	ER, PM	[46,47]
GRAMD1C	Q8IYS0	1α-C	GRAM, StART	cholesterol	cc	lipid transfer	ER, PM	[46]
GRAMD2A	Q8IUY3	1α-C	GRAM	PI(4,5)P_2_		calcium entry	ER, PM	[48]
GRAMD2B	Q96HH9	1α-C	GRAM		cc			
GRAMD4	Q6IC98	4α-N	GRAM		cc	membrane permeabilization	ER, MOM	[49]
HTRA2	O43464	1α-N	PDZ			protease		[50]
LMO7	E9PMS6	1α-N	PDZ		cc		PM, NE	[51]
MCTP1	Q6DN14	2α-C	C2 (3)				ER, SV	[52,53,54]
MCTP2	Q6DN12	2α-C	C2 (3)		cc		ER, SV	[52,53,54]
Mep1A	Q16819	1α-C	MATH		oligomer	metalloprotease	EC, PM	[55]
Miro1	Q8IXI2	1α-C	GTPase (2)		dimer	membrane transport	MOM, PO	[56,57,58]
Miro2	Q8IXI1	1α-C	GTPase (2)			membrane transport	ER, MOM, PO	[56,57,59]
MSPD2	Q8NHP6	1α-C	CRAL-Trio	phospholipid defects	cc		ER, E, G, MOM, LD	[60,61]
MYOF	Q9NZM1	1α-C	C2 (7)		cc		PM, LE, LY	[39]
ORP5	Q9H0X9	1α-C	PH	PtdSer, PI4P, PIP_2_, PIP_3_	oligomer	lipid transfer	ER, LD, PM	[62,63,64,65]
ORP8	Q9BZF1	1α-C	PH	PtdSer, PI4P, PIP_2_, PIP_3_	oligomer	lipid transfer	ER, PM, NE	[62,63,66]
OTOF	Q9HC10	1α-C	C2 (7)	PtdSer, PI(4,5)P_2_ (Ca^2+^)	cc		PM, SV	[67]
PDZD8	Q8NEN9	1α-N	SMP, PDZ, C1	PtdSer, PI4P, PI(4,5)P_2_	cc	lipid transfer	ER, LE, LY, PM	[68,69]
PLEKHB1	Q9UF11	1α-C	PH					[70]
PLEKHB2	Q96CS7	1α-C	PH	PtdSer			RE	[71,72,73]
PREB	Q9HCU5	1α-C	WD			ER export	ER, G	[74,75]
Protrudin	Q5T4F4	4α-N	FYVE	PIP_2_, PIP_3_	oligomer	lipid transfer	ER, LE	[69,76,77]
Snx13	Q9Y5W8	2α-N	RGS, PX	cholesterol, PI3P, PI(3,4)P_2_	dimer	lipid transport	EL, ER, LY	[78,79,80,81,82]
Snx14	Q9Y5W7	2α-N	RGS, PX			lipid homeostasis	ER, LD	[78,79,82,83,84]
Snx19	Q92543	2α-N	PX	PI3P			EL, ER, LD	[78,79,82]
Snx25	Q9H3E2	1α-C	RGS, PX	PIP_2_, PIP_3_	dimer	lipid homeostasis	ER, LD, PM	[78,82,85,86,87]
Spike	P0DTC2	1α-C	NTD, RBD		trimer	membrane fusion	E, EV, PM, VM	[4,5]
SYJ2B	P57105	1α-C	PDZ				MOM	
Syt1	P21579	1α-N	C2 (2)	PtdSer, PI(4,5)P_2_ (Ca^2+^)	oligomer	membrane fusion	PRCO, SV	[88,89,90,91]
Syt2	Q8N9I0	1α-N	C2 (2)	PtdSer, PI(4,5)P_2_ (Ca^2+^)	oligomer	membrane fusion	SV	[88,92,93]
Syt3	Q9BQG1	1α-N	C2 (2)	PtdSer/PI	oligomer	membrane fusion	PM, PRCO	[88,93]
Syt4	Q9H2B2	1α-N	C2 (2)	PtdSer, PI (Ca^2+^)	oligomer		SDSV, TV	[88,93]
Syt5	O00445	1α-N	C2 (2)	PtdSer, PI (Ca^2+^)	oligomer	membrane fusion	SDSV, SV	[88,93]
Syt6	Q5T7P8	1α-N	C2 (2)	PtdSer, PI (Ca^2+^)	oligomer	membrane fusion	PM, SDSV	[88,93]
Syt7	O43581	1α-N	C2 (2)	PI(4,5)P_2_ (Ca^2+^)	oligomer	membrane fusion	PM	[88,89,92,94]
Syt8	Q8NBV8	1α-N	C2 (2)		oligomer			[88]
Syt9	Q86SS6	1α-N	C2 (2)	PI(4,5)P_2_ (Ca^2+^)	oligomer	membrane fusion	SDSV, SV	[88,92]
Syt10	Q6XYQ8	1α-N	C2 (2)		oligomer	membrane fusion	TV	[88]
Syt11	Q9BT88	1α-N	C2 (2)		oligomer			[88]
Syt12	Q8IV01	1α-N	C2 (2)		oligomer		SV	[88]
Syt13	Q7L8C5	1α-N	C2 (2)					[88]
Syt14	Q8NB59	1α-N	C2 (2)		oligomer			[88]
Syt15	Q9BQS2	1α-N	C2 (2)		oligomer			[88]
TEX2	Q8IWB9	2α-N	PH, SMP	PI(4,5)P_2_		lipid transfer	ER	[68,95,96]

## Data Availability

Not applicable.

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
