# Peer review of "Transmembrane Membrane Readers form a Novel Class of Proteins That Include Peripheral Phosphoinositide Recognition Domains and Viral Spikes"

_membranes, 2022, doi:10.3390/membranes12111161_

Round 1

Reviewer 1 Report

In this manuscript, the authors define proteins that contain both peripheral region that recognize specific lipids and transmembrane region as transmembrane protein readers (TMMRs), and comprehensively review the structural characteristics of the TMMRs.  I think this review would be useful for those in the field of such proteins, thus is suitable for publication.

I have only one minor comment.  On page 2, lines 55-56, it would be useful for readership to explain that many commonly-used nanodisc systems requires solubilization by detergent, which breaks the asymmetry (, although the ref.7 is focused on the detergent-free nanodisc systems).

Author Response

Reviewer 1: Comment 1

In this manuscript, the authors define proteins that contain both peripheral region that recognize specific lipids and transmembrane region as transmembrane protein readers (TMMRs), and comprehensively review the structural characteristics of the TMMRs.  I think this review would be useful for those in the field of such proteins, thus is suitable for publication.

Authors Response:

We thank the reviewer for their assessment and recommendation of publication of our manuscript.

 Reviewer 1: Comment 2:

I have only one minor comment.  On page 2, lines 55-56, it would be useful for readership to explain that many commonly-used nanodisc systems requires solubilization by detergent, which breaks the asymmetry (, although the ref.7 is focused on the detergent-free nanodisc systems).

Authors Response: We agree that this would be useful and have altered the text as follows:

“In contrast, conventional nanodiscs formed using synthetic lipids and detergents to re-place the asymmetric biological bilayer can lead to artifacts [6], and would disrupt the native conformation of a TMMR bound to two biological membranes.”

Reviewer 2 Report

This manuscript summarizes the structural and biochemical properties of a novel class of proteins, transmembrane membrane readers (TMMRs), through the analysis of protein structure databases and structure models generated by AlphaFold and Rosetta. The authors further used MODA program to identify membrane contact sites. Overall, this manuscript fits the scope of the journal. However, there are still a few concerns that need to be addressed.

Major:

Some of the TMMRs structures are predicted by Alphafold2 and Rosetta, and further predicted the lipid binding site by MODA program. However, the accuracy of structure prediction for membrane proteins by Alphafold2 and Rosetta is relatively low, how reliable is the conclusion based on these predictions? The authors should address this uncertainty.

The authors should dock the lipids to the protein structures, at least some representatives, to show how they interact with the protein binding domains.  

Have the authors identified the differences of different lipid binding sites, for example, the difference between the PI contact sites and cholesterol contact sites?

Minor:

The last sentence in the abstract should be revised. With only one example of spike proteins, it’s not solid to make the conclusion that TMMRs represent drug discovery targets.

It is not understandable for the sentence in line 182, “identify additional membrane-binding elements N- 182 terminal to the TM helix (Fig. 2)”. Meanwhile, it is not clear what is the additional membrane-binding elements in Figure 2. It should be pointed out in the figure legends.

Minor text edits are listed below:  Please read the manuscript carefully as more errors may exist.

Line 144: add an “in” to “are found in single TMMRs”

Line 213: “play roles in Ca2+ signaling”; remove “s” in “plays” and “2+” should be superscripted.

Line 225: “catalyic” should be “catalytic”

Line 244: “a third” of what?

Author Response

Reviewer 2: Comment 1

This manuscript summarizes the structural and biochemical properties of a novel class of proteins, transmembrane membrane readers (TMMRs), through the analysis of protein structure databases and structure models generated by AlphaFold and Rosetta. The authors further used MODA program to identify membrane contact sites. Overall, this manuscript fits the scope of the journal. However, there are still a few concerns that need to be addressed.

Authors Response: We thank the reviewer for their statement that our manuscript is appropriate for the journal subject to their concerns being addressed, as we have done below.

Reviewer 2: Major Comment 2

Some of the TMMRs structures are predicted by Alphafold2 and Rosetta, and further predicted the lipid binding site by MODA program. However, the accuracy of structure prediction for membrane proteins by Alphafold2 and Rosetta is relatively low, how reliable is the conclusion based on these predictions? The authors should address this uncertainty.

Authors Response: We agree that such models can have low confidence regions such as in linkers that connect the structural domains of TMMRs. Hence we have added the following statements:

“New computational tools allow modelling of TMMRs. Full-length protein structures can be predicted from programs including AlphaFold [8] and RoseTTAFold [9]. While such artificial intelligence and machine learning methods accurately predict structures of soluble proteins [10], TMMR proteins are more challenging due to their biphasic nature, flexibly linked domains, and multimeric states. Hence, we focus here on confident-ly-predicted domains and compare models with high resolution experimental structures to define membrane-binding surfaces of TMMRs. The structures of linkers in these models are of low confidence and likely allow dynamic interactions between membranes.” As well the Figure 2 legend states: “This model contains uncertain aspects including the positioning of the two C2 domain lipid binding sites and palmitoylated cysteines, which would normally engage membrane surfaces.” Figure 7’s legend states: “This model contains uncertain aspects including the perpendicular positioning of two TM helices, the inclusion of the two predicted coiled coils in a larger all-helical domain, and the orientation of the multitude of peripheral membrane docking elements including those of the PX domain away from each other and the TM helices.”

Reviewer 2: Major Comment 3

The authors should dock the lipids to the protein structures, at least some representatives, to show how they interact with the protein binding domains. 

Authors Response:  We agree that this would be a useful addition to this review and have added Figures 6 and 8 which show the docking of PH and PX domain structures to bound lipids and micelles to illustrate how these protein binding domains engage membranes based on experimental data from NMR studies.  We also added Figure 7 which depicts an AlphaFold2 model of Snx13.

Reviewer 2: Major Comment 4

Have the authors identified the differences of different lipid binding sites, for example, the difference between the PI contact sites and cholesterol contact sites?

Authors Response: We agree that this is an interesting question worthy of further analysis, although in this review we have not identified novel differences in binding sites for such lipids. This would be a major undertaking, as currently available computational tools such as MODA does not predict lipid specificities, it only identifies membrane docking residues based on training on a set of experimentally validated phospholipid binding domains, as described in ref. 15, and the various TMMR structures and pockets are divergent and not generally well-characterized in terms of specificity. However, we do list the known lipid ligands of the various membrane binding proteins in Table 1 to assist the community in addressing this issue. We have also expanded the relevant discussion of the GRAM domains, which exhibit the best-characterized dual specificity for PIs and cholesterol: “The GRAM domain of the corresponding yeast protein, Lam6, lacks the cluster of basic residues that typically allows PH domains to recognize PIs and instead offers a more hydrophobic surface [107], suggesting a novel recognition mode.”

We also have clarified in the revised manuscript that: “Further experimental investigation is needed to determine the specificities and binding pockets of the multiple lipid ligands, as current algorithms cannot currently predict this.”

Reviewer 2: Minor Comment 5

The last sentence in the abstract should be revised. With only one example of spike proteins, it’s not solid to make the conclusion that TMMRs represent drug discovery targets.

Authors Response: We agree and have altered this sentence to say:

“The spike proteins of SARS-CoV-2 as well as other viruses are also TMMRs, seeing as they are anchored into the viral membrane while mediating fusion with host cell membranes. As such, TMMRs have key roles in cell biology and membrane trafficking, and include drug targets for diseases such as COVID-19.”

We also added the following statement just prior to the discussion to indicate wider relevance:

“Analogous mechanisms may be employed for membrane interactions by viruses including Ebola, HIV, MERS and Zika, suggesting that TMMRs may be important targets for a range of epidemics [136]”

Reviewer 2: Minor Comment 6

It is not understandable for the sentence in line 182, “identify additional membrane-binding elements N- 182 terminal to the TM helix (Fig. 2)”. Meanwhile, it is not clear what is the additional membrane-binding elements in Figure 2. It should be pointed out in the figure legends.

Authors Response: We have updated this statement to: “Our MODA analysis of the structural models derived from Rosetta and AlphaFold2 identify additional membrane-binding elements including the TVATVL-22 and ELHKIPLP-57 sequences which are N-terminal to the TM helix (Fig. 2)”

Reviewer 2: Minor Comment 7

Minor text edits are listed below:  Please read the manuscript carefully as more errors may exist.

Line 144: add an “in” to “are found in single TMMRs”

Line 213: “play roles in Ca2+ signaling”; remove “s” in “plays” and “2+” should be superscripted.

Line 225: “catalyic” should be “catalytic”

Line 244: “a third” of what?

Authors Response: We thank the reviewer for these suggested edits, which have now been made in the revised manuscript. We have also checked the rest of the manuscript and have corrected additional minor errors in the text.

Round 2

Reviewer 2 Report

The authors have now added more details, which has made this reviewer understood what was done. I do not have further questions.